# Total Variation-Based Noise Reduction Image Processing Algorithm for Confocal Laser Scanning Microscopy Applied to Activity Assessment of Early Carious Lesions

**Hee-Eun Kim [1], Seong-Hyeon Kang [2], Kyuseok Kim [3,*] and Youngjin Lee [2,*]**

[1] Department of Dental Hygiene, College of Health Science, Gachon University, 191, Hambakmoero, Yeonsu-gu, Incheon 21936, Korea; hekim@gachon.ac.kr

[2] Department of Radiological Science, College of Health Science, Gachon University, 191, Hambakmoero, Yeonsu-gu, Incheon 21936, Korea; tjdgus7345@gc.gachon.ac.kr

[3] Department of Radiation Convergence Engineering, Yonsei University, 1, Yonseidae-gil, Wonju-si, Gangwon-do 26493, Korea

\* Correspondence: seokkyu502@yonsei.ac.kr (K.K.); yj20@gachon.ac.kr (Y.L.);
Tel.: +82-10-7155-4648 (K.K.); +82-32-820-4362 (Y.L.)

**Abstract:** The confocal laser scanning microscopy (CLSM) system has been widely used to analyze early carious lesions with fluorescent ligands in dental imaging. This system can be used to examine the physiological condition of cellular colonization in the tooth structure. However, the undesirable noise in CLSM images hinders accurate activity assessment of early carious lesions. To address this limitation, a total variation (TV)-based noise reduction algorithm with good edge preservation was developed, and its applicability to medical tooth specimen images obtained with CLSM was verified. To evaluate the imaging performance, the proposed algorithm was compared with conventional filtering methods in terms of the normalized noise power spectrum, contrast-to-noise ratio, and coefficient of variation. The results indicate that the proposed algorithm achieved better noise performance and fine-detail preservation, in comparison with the conventional methods.

**Keywords:** total variation noise reduction algorithm; confocal laser scanning microscopy; medical diagnosis; caries activity; evaluation of image performance

---

## 1. Introduction

In the fields of biomedicine and microbiology, fluorescence microscopy is employed to image proteins, tissues, and cells, which are otherwise impossible to observe with the naked eye. The expression and location of a particular protein inside a living cell can be analyzed in real time using fluorophores, which are fluorescent chemical compounds that can re-emit light on light excitation. However, it is difficult to accurately image a sample using a conventional fluorescence microscope. Hence, confocal laser scanning microscopy (CLSM), which can achieve a higher resolution than that of a conventional fluorescence microscope, was introduced. Unlike the conventional approach, the main/underlying principle of CLSM involves passing a laser beam through a light source aperture and subsequently focusing it on the sample surface using an objective lens. A confocal laser scanning micrograph is constructed pixel-by-pixel by collecting the emitted photons from the fluorophores in the sample [1,2]. Based on this basic principle, CLSM provides images with higher resolution (approximately 1.4×). In addition, thick specimens can be imaged via optical sectioning, while autofluorescence from outside the focal plane can be eliminated using spatial filtering. Furthermore,

spatial depth control is feasible through optical slicing. Wang et al., employed an interference confocal microscope with a simultaneous phase-shifter device to achieve an axial depth resolution of 1 nm [3]. CLSM can be used to image hard tissues, which are more difficult to capture compared to soft tissues. This approach has been employed in the field of dentistry [4].

CLSM has been used to identify the activity of early carious lesions [5–7]. Schupbach et al. observed a higher concentration of bacteria in active lesions than in arrested lesions [8]. However, simple microbiological tests (for, e.g., colony forming unit assays) are not useful for differentiating between active and arrested dental caries. Using CLSM to accurately observe live bacteria within the early carious lesions on the enamel can contribute to the development of diagnostic criteria for early caries based on the lesion activity. In particular, F. G. de Carvalho et al., used CLSM to measure the bactericidal effect against a biofilm in the tooth from 20 to 590 s and confirmed that the antibacterial effect improved when Protect Bond was used [9]. In addition, N. Ciacotch et al., verified that a copper-silver alloy can achieve low bacterial contamination using a tailor-made CLSM to visualize the killing of bacterial biofilms [10].

The resolution of a CLSM image is poor owing to the inevitable noise generated while processing the contrast. Image noise can limit the early detection of carious lesions in tooth specimen images. In addition, the edge information is low owing to the limited amount of light emanating from the focused beam. Total variation (TV)-based approaches, which were first introduced by Rudin et al., in 1992, have been applied to medical systems to reduce image noise while preserving edge information [11]. Many studies have been conducted to evaluate the applicability of TV-based noise reduction algorithms to various medical imaging systems [12–16]. Wilson et al., applied an optimization model with a TV-based noise reduction algorithm to a chest computed radiography (CR) X-ray image and reported a sensitivity of approximately 79% [12]. In addition, our previous research studies reported that the TV-based noise reduction algorithm outperformed conventional filters, such as the median and Wiener filters, when applied to computed tomography (CT) and chest X-ray medical imaging systems [13,15]. However, studies on the applicability of this algorithm to CLSM images are limited. In this context, a TV-based noise reduction algorithm for CLSM images was developed, and its image performance was evaluated in terms of the normalized noise power spectrum (NNPS), contrast-to-noise ratio (CNR), and coefficient of variation (COV).

## 2. Materials and Methods

### 2.1. Sample Collection and Ethical Considerations

This study was approved by the Institutional Review Board (IRB) committee of Gachon University (IRB No. 1044396-201711-HR-185-01). All procedures were conducted in accordance with the ethical principles for medical research involving human participants, as stipulated in the Declaration of Helsinki (2013 version) by the World Medical Association. The participants were recruited between November 2017 and March 2018. The objective and procedures of the study were explained to all participants. Prior to tooth extraction, which was necessary in the case of periodontal disease or orthodontic treatments, written informed consent for the use of the extracted teeth in the proposed research was obtained from all the participants. Following tooth extraction, any attached soft tissue was carefully removed using a toothbrush and dental explorer. The teeth were stored in a brain heart infusion broth (BHI; BD Difco, Sparks, MD, USA) at 37 °C until use.

### 2.2. CLSM Analysis

Prior to performing CLSM, the collected human teeth were screened using a quantitative light-induced fluorescence-digital system (QLF-D Biluminator™2+, Inspektor Research Systems BV, Amsterdam, the Netherlands) to detect early carious lesions. The QLF-D examinations were conducted in a controlled and dark environment; the distance between the camera lens and teeth was maintained throughout the process.

The human enamel specimens were cut perpendicular to the superficial surfaces (Figure 1). A 300-µm-thick section was cut (TechCut 4, Allied High Tech Products, Inc., Rancho Dominguez, CA, USA) from each specimen (Figure 1) and grounded to a thickness of 100 µm with abrasive papers of 1200 grit (SiC paper, R&B Inc., Daejeon, Republic of Korea). The sample thickness was measured using a digital micrometer (T-M039 Exploit®, Tokyo, Japan) with a precision of 0.001 mm.

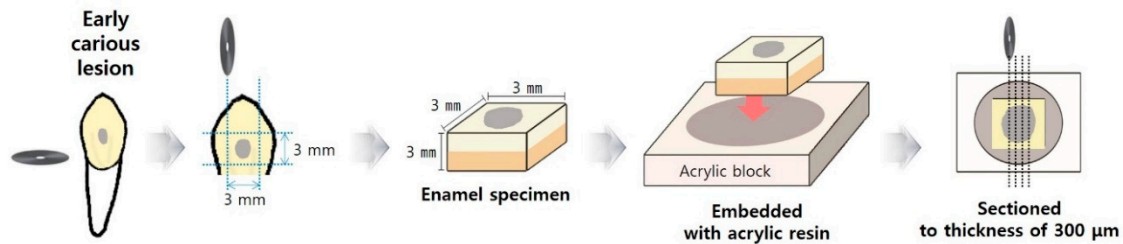

**Figure 1.** Schematic diagram illustrating the sample preparation for confocal laser scanning microscopy (CLSM) imaging.

To stain the early carious lesion area, the sectioned specimens were mounted on glass slides and dyed with 10 µL of SYTO-9 and propidium iodide (LIVE/DEAD® BacLight™ Bacterial Viability Kit, Thermo Fisher Scientific, Massachusetts, USA) staining reagents using a pipette. The dyed specimens were subsequently incubated at room temperature in the dark for 15 min to obtain red and green fluorescent images. Subsequently, the specimens were observed under a confocal laser scanning microscope (Zeiss LSM 510, Carl Zeiss MicroImaging GmbH, Jena, Germany) equipped with a He–Ne laser at an excitation wavelength of 543 nm. The specimens were imaged at a magnification of 50× in a dark room.

### 2.3. Modeling of the Total Variation (TV)-Based Noise Reduction Algorithm

A TV-based noise reduction algorithm that can reduce different types of noise in medical images while preserving edge information was designed.

The general equation for the image degradation process with noise components can be expressed as:

$$\text{A}(x,\ y) = h(x,y) ** PSF(x,y) + N(x,y)\ (** : 2D\ convolution) \tag{1}$$

were, $\text{A}(x,y)$ is the degraded image; $h(x,y)$ is the original image; $PSF(x,y)$ is the shift-invariant point spread function; and $N(x,y)$ is the noise. A regularization function is generally used to solve the unknown term $h(x,y)$. Based on the image degradation process, the proposed algorithm can be expressed as [11]:

$$V_{TV}[u_s] = \int_{\Omega} |\nabla u_s| dx dy\ \left(\nabla u_s = \sqrt{u_{s_x}{}^2 + u_{s_y}{}^2}\right) \tag{2}$$

where, $V_{TV}$ is the total variation; $\nabla u_s$ is the gradient; $u_{s_x}$ and $u_{s_y}$ are $\frac{\partial u_s}{\partial x}$ and $\frac{\partial u_s}{\partial y}$, respectively.

To compare the image performances of different noise reduction algorithms, conventional methods incorporating the median and Wiener filters were designed.

*2.4. Evaluation of Image Performance (Noise Characteristics)*

The performance of the designed algorithm for CLSM was evaluated in terms of the NNPS, CNR, and COV, which are defined as

$$
\begin{aligned}
\text{NNPS}(u,v) &= \frac{\text{NPS}(u,v)}{(\text{mean signal of average ROI})^2}\\
(\text{NPS}(u_n,v_k) &= \lim_{N_x,N_y \to \infty} (N_x N_y \Delta x \Delta y) < |FT_{nk}I(x,y) - S(x,y)|^2 >\\
&= \lim_{N_x N_y \to \infty} \lim_{M \to \infty} \frac{(N_x N_y \Delta x \Delta y)}{M} \sum_{m=1}^{M} |FT_{nk}I(x,y) - S(x,y)|^2\\
&= \lim_{N_x N_y, M \to \infty} \frac{\Delta x \Delta y}{M \cdot N_x N_y} \sum_{m=1}^{M} < | \sum_{i=1}^{N_x} \sum_{j=1}^{N_y} (I(x_i,y_j) - S(x,y)) \exp(-2\pi i(u_n x_i + v_k y_i))|^2)
\end{aligned}
\tag{3}
$$

$$
\text{CNR} = \frac{|O_T - O_B|}{\sqrt{\sigma_T{}^2 + \sigma_B{}^2}}
\tag{4}
$$

$$
\text{COV} = \frac{\sigma_T}{O_T}
\tag{5}
$$

where, $u$ and $v$ are the spatial frequency conjugates in the X and Y directions, respectively; $N_x$ and $N_y$ are the number of pixels in the X and Y directions, respectively; $\Delta x$ and $\Delta y$ are the pixel spacings in the X and Y directions, respectively; $I(x_i, y_j)$ is the image intensity at the $(x_i, y_j)$ pixel location; $S(x,y)$ is the mean intensity; $O_T$ and $\sigma_T$ are the mean and standard deviation, respectively, of the target region of interest (ROI); $O_B$ and $\sigma_B$ are the mean and standard deviation, respectively, of the background ROI. The established/selected ROIs represent the early carious lesions in the acquired image.

## 3. Results

We investigated the feasibility of the proposed algorithm for CLSM images when applied for the activity assessment of early carious lesions. Figure 2 shows the acquired images of the tooth specimens including the CLSM image with ROIs for calculating the NNPS, CNR, and COV. Figure 3 shows the images obtained with the conventional noise reduction filters and the proposed algorithm. Based on the magnified region in the CLSM images, it can be observed that there is reduced noise in the images obtained with the proposed algorithm.

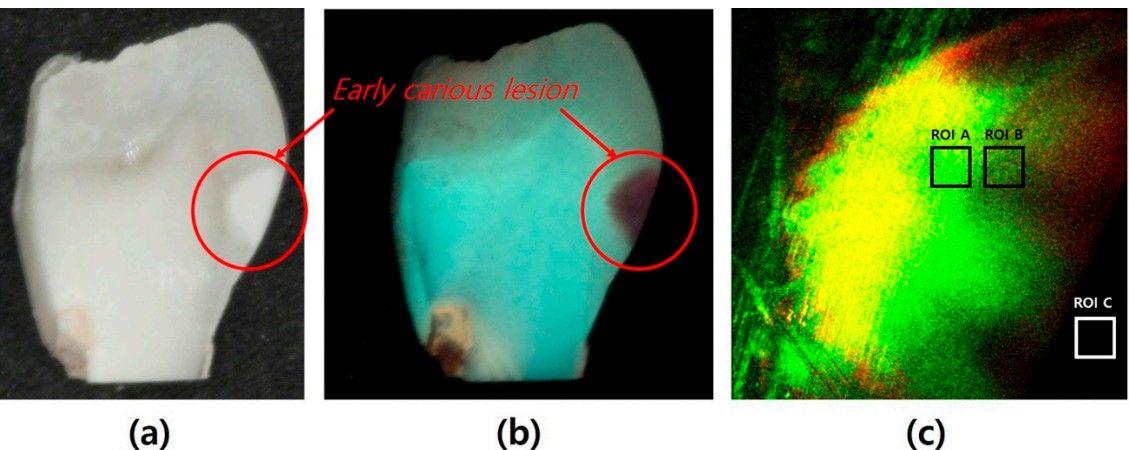

**(a)**　　　　**(b)**　　　　**(c)**

**Figure 2.** (**a**) White-light image of the tooth specimen obtained using quantitative light-induced fluorescence-digital system (QLF-D), (**b**) fluorescence image of the tooth specimen obtained with QLF-D, and (**c**) CLSM images of the tooth specimen showing region of interest (ROI) A (contrast-to-noise ratio (CNR), and coefficient of variation (COV) calculations), ROI B (CNR calculations), and ROI C (normalized noise power spectrum (NNPS) calculations).

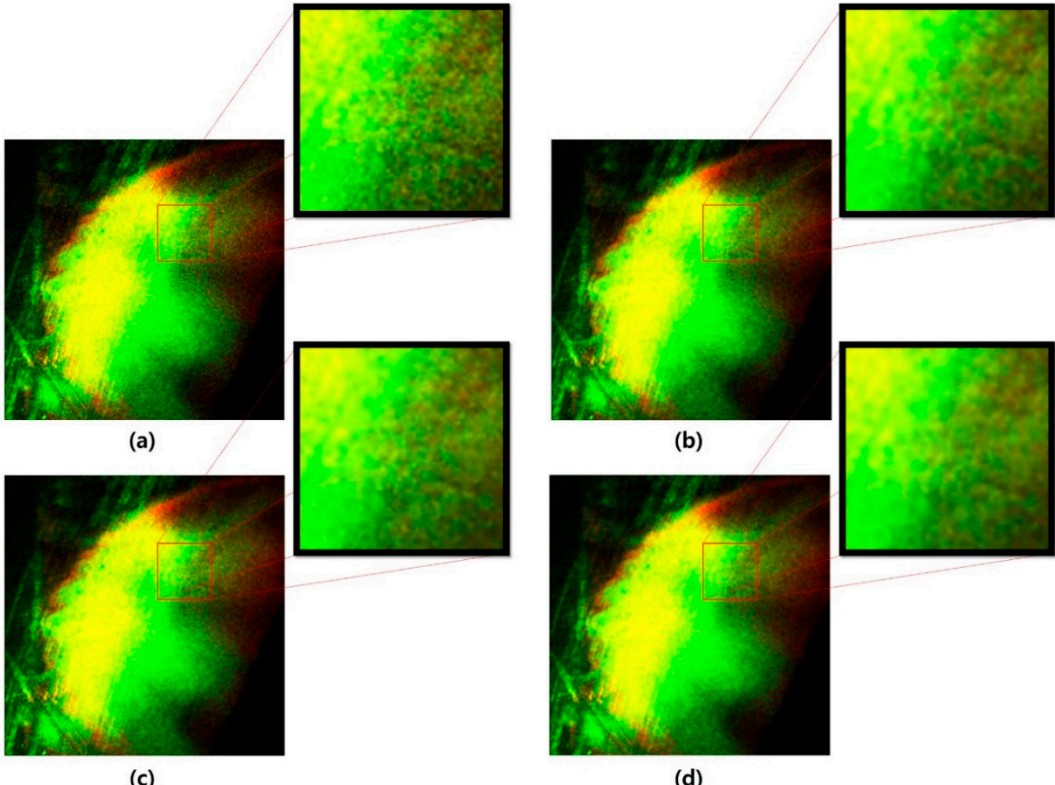

**Figure 3.** Tooth specimen obtained using CLSM, including the magnified region: (**a**) noisy image, (**b**) conventional median filter, (**c**) conventional Wiener filter, and (**d**) proposed total variation (TV) noise reduction algorithm.

Accurate and early detection of carious lesions from CLSM images is difficult because of additive noise, which is often characterized using the noise power spectrum (a meaningful parameter used to compare different imaging methods or systems). Figure 4 presents the NNPS results for the conventional noise reduction filters and proposed algorithm.

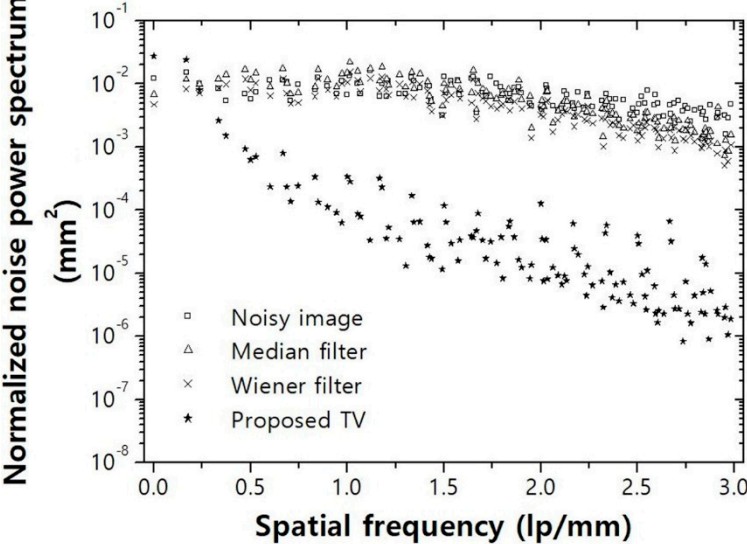

**Figure 4.** NNPS results (over ROI C shown in Figure 2) obtained for the noisy image, median filter, Wiener filter, and proposed TV noise reduction algorithm applied to the confocal laser scanning micrograph of the tooth specimen.

Figure 5 presents the evaluated CNR and COV results for the conventional noise reduction filters and proposed algorithm. The proposed algorithm achieved the highest CNR, with the CNR of the noisy image < median filter < Wiener filter < proposed algorithm. In addition, the proposed algorithm also exhibited the lowest COV, with the COV of the proposed TV noise reduction algorithm < conventional Wiener filter < conventional median filter < noisy image.

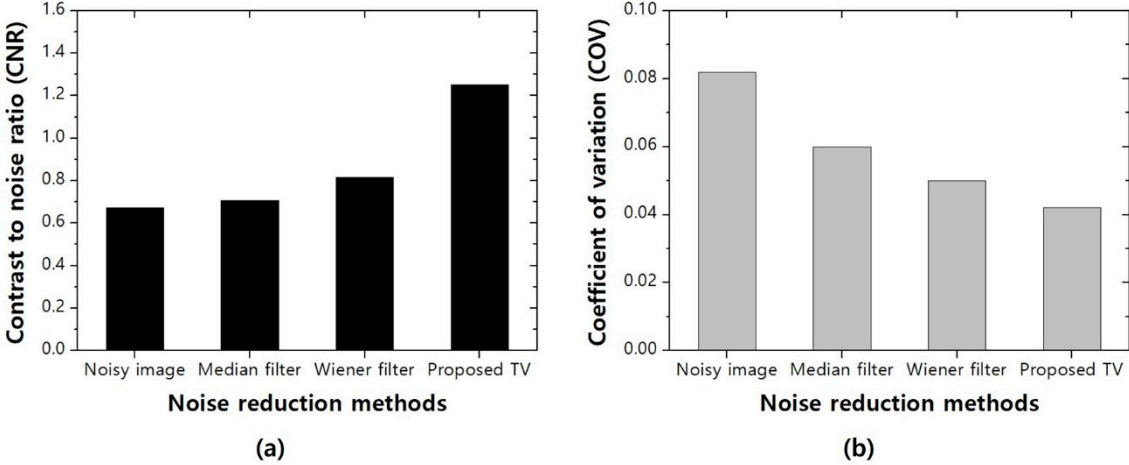

**Figure 5.** (**a**) CNR and (**b**) COV results (over ROI A and ROI B shown in Figure 2) obtained for the noisy image, median filter, Wiener filter, and proposed TV noise reduction algorithm applied to the confocal laser scanning micrograph of the tooth specimen.

## 4. Discussion

Conventional light microscopy is limited by the achievable image quality, in terms of the resolution and noise performance, owing to fundamental factors. The CLSM imaging system has superior transverse resolution in the direction parallel to the X and Y-axes and improved longitudinal resolution in the direction parallel to the Z-axis on the specimen, compared to a conventional microscope with the same objective lens at the same wavelength [17]. Even in a CLSM imaging system with improved resolution, noise removal is essential and must be efficiently improved to facilitate the early detection of carious lesions. In this study, a TV-based noise reduction algorithm was developed to achieve improved noise removal, facilitating the diagnosis of carious lesions with improved efficiency.

The TV noise reduction algorithm had the lowest NNPS value (approximately $10^{-6}$ mm$^2$) in the spatial frequency range, which was $10^4$ times lower than those of the median and Wiener filters (Figure 4). Therefore, this confirmed that the proposed algorithm could achieve a considerably greater degree of noise reduction/removal and provide improved noise characteristics. In addition, the NNPS through random selection was smoother. In the case of the proposed algorithm, the NNPS value decreased rapidly in the spatial frequency range of approximately 0.2–1.0 lp/mm and subsequently decreased gradually until reaching the Nyquist frequency (Figure 4).

The CNR of the proposed algorithm was approximately 1.86, 1.77, and 1.53 times higher than that of the noisy image, median filter, and Wiener filter, respectively (Figure 5). The COV of the proposed algorithm was approximately 1.90, 1.40, and 1.29 times better than that of the noisy image, median filter, and Wiener filter, respectively (Figure 5).

P. Hanninen et al., reported on the applicability of conventional filtering methods using a non-linear filter [18], whereas S. C. Lee et al., suggested an intensity correction method with mean filtering following contrast limiting adaptive histogram equalization for CLSM images [19,20]. However, while non-linear filters have been used since long, these filters have the disadvantage of a low denoising efficiency. In this study, for the first time, an iterative TV noise reduction algorithm has been applied to CLSM images to improve the denoising efficiency.

The quantitative evaluation results of the noise reduction methods applied to the CLSM imaging system reveal that the designed algorithm achieved excellent image quality. The noise in the CLSM image was more efficiently reduced, in comparison with the conventional filters, owing to the use of denoising models that reject Gaussian and Poisson noises and a smoothing function. The improved results reflect the efficiency of the proposed algorithm for noise reduction by using TV to model the image intensity function, solving the constrained optimization problem using Lagrange multipliers and regularizing the coefficient functions. The results verify that the designed algorithm can help reduce noise in medical tooth specimen images captured with CLSM.

To compare the performances in terms of the noise and spatial resolution, the modulation transfer function (MTF) was calculated owing to the trade-off between the noise and spatial resolution. The MTFs corresponding to the different image processing algorithms used for noise reduction were measured. The blind deconvolution method, i.e., the deconvolution method to identify the unknown $PSF(x, y)$ from microscopy images, was employed. Briefly, the estimated $psf^{(i)}$ can be efficiently computed by solving Equation (6) using the conjugate-gradient method as:

$$\nabla^2 A = \nabla^2 h^{(i)} \otimes \otimes PSF^{(i)} \tag{6}$$

where $A$ is the degradation image, $\nabla^2$ is the 2nd order differential operator used for edge detection in images, and $h^{(i)}$ is computed and updated by following the object function ($\varnothing(h^{(i)})$) as defined in Equation (7).

$$\varnothing(h^{(i)}) = \underset{h^{(i)} \in Q}{\operatorname{argmin}} \frac{\mu}{2} \|Bh^{(i)} - A\|_2^2 + \|\nabla h^{(i)}\|_1 \tag{7}$$

Here, $B$ is the convolution matrix, and $\mu$ is a regularization parameter. The object function ($\varnothing(h^{(i)})$) can be solved using the augmented Lagrangian method. The estimated $h^{(i)}$ is reused in Equation (7), and the optimized $PSF(x, y)$ is obtained. Figure 6 details the calculated MTF. Although the noisy image exhibited superior performance in terms of the MTF, similar data were acquired for all the noise reduction methods including the noisy image.

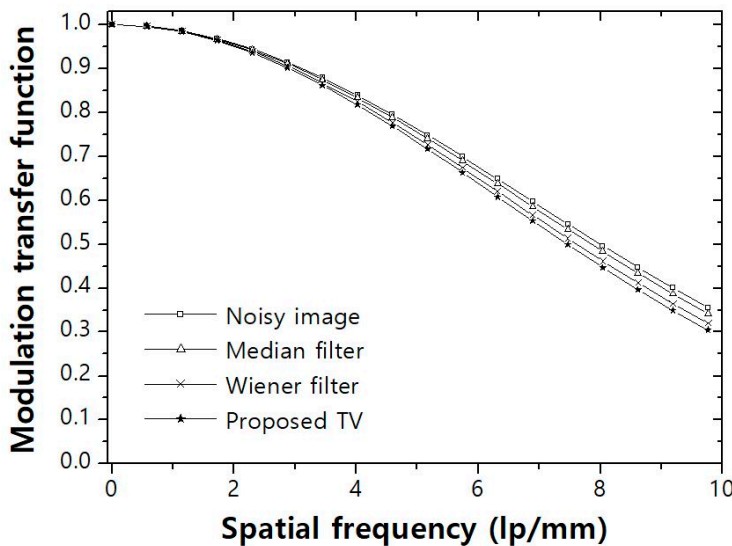

**Figure 6.** Modulation transfer function (MTF) results obtained for the noisy image, median filter, Wiener filter, and proposed TV noise reduction algorithm applied to the confocal laser scanning micrograph of the tooth specimen.

## 5. Conclusions

In this study, experiments were conducted to evaluate the efficiency of a TV-based noise reduction algorithm for CLSM images. The results indicated that the proposed algorithm achieved good image

quality in terms of the NNPS, CNR, and COV for assessing the activity of carious lesions in the field of cariology.

The two color channels of a CLSM image consistently contain different amounts of noise owing to multiple sensitivities. Therefore, future studies will focus on modifying the proposed algorithm and evaluating the noise reduction performance using various color channels based on the CLSM imaging system.

**Author Contributions:** Conceptualization, H.-E.K. and Y.L.; data curation, S.-H.K., K.K. and Y.L.; formal analysis, K.K. and Y.L.; funding acquisition, H.-E.K. and Y.L.; software, S.-H.K. and K.K.; writing—original draft, H.-E.K. and S.-H. K.; writing—review and editing, K.K. and Y.L. All authors have read and agreed to the published version of the manuscript.

**Funding:** This research was supported by the National Research Foundation of Korea (NRF) funded by the Ministry of Science and ICT (No. NRF-2019R1F1A1062811 & NRF-2019R1F1A1058152).

**Conflicts of Interest:** "The authors declare no conflicts of interest."

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
