# Peer review of "Total Variation-Based Noise Reduction Image Processing Algorithm for Confocal Laser Scanning Microscopy Applied to Activity Assessment of Early Carious Lesions"

_applsci, doi:10.3390/app10124090_

Round 1
Reviewer 1 Report
In this study, authors developed a total variation (TV)-based noise reduction algorithm with fine edge preservation and confirmed its applicability to medical tooth specimen images obtained using confocal laser scanning microscopy (CLSM). To evaluate the performance of the proposed method, the authors compared the proposed algorithm with conventional filtering methods in terms of the normalized noise power spectrum, contrast-to-noise ratio, and coefficient of variation.
This manuscript is well written, and the objective is clear. There are enough analyses to support the claim.
I recommend the publication of this manuscript as is.
Author Response
√ Thank you for review and comments.
Reviewer 2 Report
It is a original work that presents new algorithm for improvement of resolution of confocal laser scanning microscopic systems. Authors proposed to apply developed approach to activity assessment of early carious lesions.
No doubts, this approach is correct, effective and interesting from scientific point of view. However, there are some notes, which authors must take into consideration:
- The practicability of use of confocal laser scanning microscopy in dental medicine is not clear. Authors carried out their experiments using extracted human tooths, but they did`t explain how such systems can be used for study of bacteriological activity in tooths with early carious lessions of real patients ?
If confocal laser scanning microscopy is really used for such applications, it is necessary to cite corresponding papers.
- Authors did`t explain which resolution of optical imaging systems is required for accurate early detection of carious lesions. They tried to improve resolution confocal laser scanning microscopic systems and achieved good results, but it is absolutely not clear how analysis of carious lesions is conducted in conventional dental practice and why resolution of existing systems is not enough ?
To summarise: This research work is worth publication, but it needs some corrections. Authors have to answer on above questions and prepare more detailed overview of scientific literature.
Author Response
Reviewer 2:
Dear reviewer,
Thank you for your useful comments and suggestions concerning our paper entitled “Total variation-based noise reduction image processing algorithm for confocal laser scanning microscopy applied to activity assessment of early carious lesions”.
It is a original work that presents new algorithm for improvement of resolution of confocal laser scanning microscopic systems. Authors proposed to apply developed approach to activity assessment of early carious lesions.
No doubts, this approach is correct, effective and interesting from scientific point of view. However, there are some notes, which authors must take into consideration:
- The practicability of use of confocal laser scanning microscopy in dental medicine is not clear. Authors carried out their experiments using extracted human tooths, but they did`t explain how such systems can be used for study of bacteriological activity in tooths with early carious lessions of real patients? If confocal laser scanning microscopy is really used for such applications, it is necessary to cite corresponding papers.
√ Thank you for variable comments. As you commented, we added sentences and reference for explain how CLSM system can be used for study of bacteriological activity in tooth.
- Added sentences in ‘Introduction section’: In particular, F. G. de Carvalho et al. studied that the CLSM was used to measure the bactericidal effect against biofilm in tooth from 20 to 590 seconds, and it was confirmed that the antibacterial effect was improved when Protect Bond was used [9]. In addition, N. Ciacotch et al. demonstrated that copper-silver alloy can achieve low bacterial contamination by using a tailor-made CLSM to visualize the killing of bacterial biofilms [10].
- Added references:
[9] F. G. de Carvalho, R. M. Puppin-Rontani, S. B. P. de Fucio, T. C. Negrini, H. L. Carlo, F. Garcia-Godoy, Journal of Applied Oral Science 2012, 20, 568-575.
[10] N. ciacotich, K. N. Kragh, M. Lichtenberg, J. E. Tesdorpf, T. Bjarnsholt, L. Gram, Global Challenges 2019, 3, doi: 10.1002/gch2.201900044.
- Authors did`t explain which resolution of optical imaging systems is required for accurate early detection of carious lesions. They tried to improve resolution confocal laser scanning microscopic systems and achieved good results, but it is absolutely not clear how analysis of carious lesions is conducted in conventional dental practice and why resolution of existing systems is not enough ?
√ Thank you for variable comments. We discussed the effects on noise on early detection of carious lesions and a comparative explanation of the existing methods.
- Added sentences in ‘Discussion section’: The conventional light microscopy is limited in the image quality such as resolution and noise performance due to fundamental factors. The CLSM imaging system has superior transverse resolution in the direction parallel to the X and Y axes on the specimen surface and improved longitudinal resolution in the direction parallel to the Z axis on the specimen, compared to a conventional microscope using the same objective lens at the same wavelength [17]. Even in the CLSM imaging system with improved resolution, noise is essential and must be efficiently improved for early detection carious lesions. In this study, we tried to improve efficiency of carious lesion diagnosis by improving noise by applying an efficient TV-based noise reduction algorithm.
- Added reference:
[17] P. M. Buscemi, Clinical Experience With the ConfoScan 2 Corneal Confocal Microscope, Journal of Refractive Surgery 2000, 2, S276-S280.
To summarise: This research work is worth publication, but it needs some corrections. Authors have to answer on above questions and prepare more detailed overview of scientific literature
√ Thank you for variable comments.

Reviewer 3 Report
The Authors present a well-known noise reduction method useful for many imaging techniques as an improvement for CLSM applications.
The attempt can be interesting, but their effort is hampered by the fact that several previous studies and results exist that are not correctly cited (see the statement at line 66).
In my opinion, a better control of the available literature, together with a clear statement about the novelty of the presented paper, is mandatory because the only adoption of TV denoising procedures for better CLSM images is not enough for a publication. More application examples and a specific noise modelling for the CLSM could be also desirable.
There is a typo at line 45: the obtained depth resolution by that work was 1 nm (vs. 1 mm).
Author Response
Reviewer 3:
Dear reviewer,
Thank you for your useful comments and suggestions concerning our paper entitled “Total variation-based noise reduction image processing algorithm for confocal laser scanning microscopy applied to activity assessment of early carious lesions”.
The Authors present a well-known noise reduction method useful for many imaging techniques as an improvement for CLSM applications.
The attempt can be interesting, but their effort is hampered by the fact that several previous studies and results exist that are not correctly cited (see the statement at line 66).
√ Thank you for comment. I’m sorry confusion you. We modified reference for citation in our manuscript (from “[11-14]” to “[13, 15])
In my opinion, a better control of the available literature, together with a clear statement about the novelty of the presented paper, is mandatory because the only adoption of TV denoising procedures for better CLSM images is not enough for a publication. More application examples and a specific noise modelling for the CLSM could be also desirable.
√ Thank you for variable comments. We improved sentences for TA denoising procedures for better CLSM images as your recommendation.
- Added sentences in “Discussion section”: P. Hanninen et al. showed application feasibility of conventional filtering methods using non-linear filter [18] and S. C. Lee et al. suggested intensity correction method with mean filtering method following the contrast limiting adaptive histogram equalization method in CLSM image [19, 20]. However, non-linear filters have been used for a long time and have a disadvantage of low denoising efficiency. In this study, we tried to improve efficiency by applying TV noise reduction algorithm based on the approach to iterative-based solution method to CLSM images for the first time.
- Added references:
[18] P. Hanninen, E. H. K. Stelzer, J. Salo, Nonlinear Filtering in Improving the Image Quality of Confocal Fluorescent Images 1991, 4, 243-253.
[19] E. D. Pisano, S. Zong, B. M. Hemminger, M. DeLuca, R. E. Johnston, K. Muller, M. P. Braeuning, S. M. Pizer, Contrast Limited Adaptive Histogram Equalization Image Processing to Improve the Detection of Simulated Spiculations in Dense Mammograms, Journal of Digital Imaging 1998, 11, 193-200.
[20] S. C. Lee, P. Bajcsy, Intensity correction of fluorescent confocal laser scanning microscope images by mean-weight filtering, Journal of Microscopy 2006, 221, 122-136.
There is a typo at line 45: the obtained depth resolution by that work was 1 nm (vs. 1 mm).
√ Thank you for comment. We revised sentence as your comment (from “1 mm” to “1 nm”).
